# Multiple Steady States in the Photocatalytic Reactor for Colored Compounds Degradation

**DOI:** 10.3390/molecules26133804

**Published:** 2021-06-22

**Authors:** Jakub Szyman

**Affiliations:** Department of Chemical Engineering and Technology, Cracow University of Technology, Warszawska 24, 31-155 Krakow, Poland; jakub.szyman@pk.edu.pl; Tel.: +48-126282730

**Keywords:** multiple steady states, photocatalytic reactor, wastewater treatment, colored compounds

## Abstract

The paper reports the occurrence of multiple steady-state zones in most of the constructions of fixed-bed photocatalytic reactors. Such a phenomenon has not been ever observed in a field of photocatalytic reactors. The simulation has been provided for a common case in a photocatalysis—the degradation of colored compounds. The mathematical model of the photocatalytic reactor with immobilized bed has been stated by a simple ideal mixing model (analogous to the CSTR model). The solution has been continued by the two parameters—the Damköhler number and the absorption coefficient related to the inlet stream concentration. Some branches of steady states include the limit point. The performed two-parametric continuation of the limit point showed the cusp bifurcation point. Besides the numerical simulation, the physical explanation of the observed phenomenon has been provided; the multiple steady-states occurrence is controlled by light absorption–reaction rate junction. When the reaction rate is limited by the light absorption, we can say that a *light barrier occurs*. The dynamical simulations show that when the process is operated in a field of multiple steady states, the overall reactor efficiency is related to the reactor set-up mode.

## 1. Introduction

Photocatalytic wastewater treatment is developed as a low-cost, environmentally friendly process which provides non-selective oxidation of water impurities. Numbers of experimental works are managed in the case of colored compounds [1,2,3]. Textile [4,5], leather [6], agricultural [7], dairy [8] and many more industry branches may become the potential benefiters of the development of the photocatalytic wastewater treatment technology. However, there are still scientific challenges in transferring the technology from laboratory to industry scale [9,10]. One of them is combined with the economical factors. Each new apparatus in the technological chain generates additional process costs. Therefore, the photocatalytic reactor construction should provide possible low operation costs coupled with possibly maximal process efficiency. Using the immobilized-bed catalyst prevents the loss of the high-cost catalyst during the process. The simplest case of such a reactor is the thin film fixed-bed photocatalytic reactor (TFFBPR), which is a sloped plate or an rectangular tank with the catalyst surface on its bottom wall [11,12]. Nevertheless, the design of the reactor, which is dedicated to the particular process, is more complex. In such a case, the designer should provide the stable operation conditions.

The multiple steady states are one of the phenomena which is present in many cases of chemical reactors and it can affect the process efficiency. Processes in fluidized-bed reactors [13] and bioprocesses [14,15] are only examples. Such a phenomenon is related to the nonlinearity of the mathematical model. In the case of chemical reactors, the nonlinearity is usually integrated into the kinetic model. Other papers showed that the multiple steady states structure may be affected by inlet and outlet streams operation set-up [16], the mechanics of the set of elementary processes [17] or other physicochemical factors determining the form of the kinetic equation and the relation between the processes of transferring the quantities, which are integrated in the kinetic equations. The very last case may be illustrated by the adiabatic reactor, where the multiple steady states may be related to both hydrodynamics of the reactor and the heat exchanger, which provides the proper heat transfer conditions [18]. On the other hand, the problem is more complex in the case of multiphase heterogeneous catalysis in which such processes should be considered in every phase taking into the account also the mass transfer between them. Photocatalysis is a heterogeneous process that is additionally related to the light incident on the catalyst surface. For such many reasons, it is expected that multiple steady states may occur in photocatalytic reactors.

In this paper, I showed the separated case which is related to the interaction between the partial radiation absorption by the solution of the degraded compound and the photocatalytic reaction rate. It is obvious that the photocatalytic process rate in some conditions may be combined with the light intensity on the catalyst surface. The light incidence partially decreases when the degraded compound absorbs the light. In such a case, it is expected that the reaction rate will decrease. On the other hand, the degradation of water contaminants is leading to decreasing its concentration. In consequence, more light is reaching the catalyst surface. In specific circumstances, the light absorption—the degradation rate junction—may cause the multiple steady-states occurrence. This research is about the more general case of such a junction. This is the first part of this study. On the other hand, every chemical reactor is related to the hydrodynamical conditions. Therefore, in the second part I presented the simple multiple steady-states analysis where the liquid stream inside the reactor is only partially mixed in the axial direction. This simulation is simple, in order to clearly illustrate that such a phenomenon may be applied in the real case.

## 2. Mathematical Model

The research has been realized by numerical simulations. To set it up, the mathematical model is needed. The model is stated by the mass balance of the degraded substrate in the reactor. Photocatalytic reactors, in general, can be described by the heterogenic reactor model. Therefore, the mass balance in such a case is performed both for the liquid phase and the catalyst surface. The photocatalytic reactors cannot be treated as in a classical heterogeneous catalysis problem. The mass transport inside the photocatalytic bed practically does not affect the overall process rate, because the photons absorbed inside the bed recombine orders of magnitude quicker than the reactant we transferred there. In my previous paper, we figured out that, in the case of immobilized heterogeneous photocatalysis, the reaction is taking part only on the external catalytic layer [19]. Therefore, in general, the model of the photocatalytic reactor for degradation of a wastewater contaminant consists of:(1)the substrate balance in the liquid phase;(2)the substrate balance on the (external) catalyst surface.

In the investigated case, the catalyst is immobilized on the bottom wall of the reactor. Such an assumption is referred to the construction of many reactors, both the industrial [20] and the laboratory scale [5,21].

## 2.1. Kinetic Model

In this paper, the kinetics of the photocatalytic process are defined by the commonly used Langmuir–Hinshelwood model.
r_A_ = k_L−H_ Kc_A_/(1 + Kc_A_),(1)

In most of research, the kinetics are pseudo-first-order; this is because the concentration of degraded compounds in the sewage is usually small to provide the pseudo-first-order kinetics. When c_A_ is so small, then the expression in the denominator of Equation (1) goes to 1. Therefore, the reaction rate can be rewritten for such a case:r_A_ = k_L−H_ Kc_A_,(2)

Some authors combine the Langmuir–Hinshelwood constant with the molar photon flux and the quantum yield [22,23]. The quantum yield may be defined in many ways. The modeling of the photocatalytic reactor does not require taking into account the detailed kinetics mechanics. It is shown that concentrations in the reactor volume of most of the intermediates between the degraded contaminate and finally its mineralization is negligible. In the field of interest for photocatalytic reactor modeling is how much of the photons incoming to the catalyst surface can cause the degradation of one molecule of absorbed contaminant. On the other hand, it is more comfortable to express the numbers of photons and molecules in moles. Finally, the quantum yield used in this paper is presented by Equation (3):φ = (moles of degraded contaminant)/(moles of absorbed photons),(3)

The incorporation of the quantum yield and the molar photon flux absorbed by the catalyst is leading to the form of the kinetic Equation (4):r_A_= φI_a_Kc_A_,(4)

Before the radiation comes to the catalyst surface, it has to penetrate the liquid layer. Thus, I have to take into an account the partial light absorption by the water contaminant. The pathway of the light ray through the liquid is influenced by many physical conditions; the most important are: the shape of the liquid volume and the method of the light emission by the light source. For example, the standard high-pressure mercury lamp emits the light beam cylindrically, whereas the sunlight is falling perpendicularly to the zenith angle. Instead of that, the light from the LED diode is outcoming conically from the single point. On the other hand, the shape of the reactor vessel determinates the pathway of the light to the catalyst surface. In this paper, I performed the easiest-to-model case; the light irradiates the reactor vertically, whereas the reactor has a cuboid shape. Such an assumption leading to the case is described by the Lambert–Beer law. The light incoming to the catalyst surface is partially absorbed by the layer of the reacting solution and it can be calculated from Equations (5) and (6):dI/dy = Iεc_A_(y),(5)
I(y = δ) = I_0_(6)
where d is a thickness of the liquid layer in the cuboid reactor.

The partial light absorption by the reacting solution is related to the water contaminant concentration. The function of the concentration with respect to the vertical coordinate c_A_(y) is related to the transverse diffusion of the contaminant to the catalyst surface. In my previous paper, we found that there are some limiting conditions for which the transverse mass dispersion is not a limiting factor for the overall reactor efficiency. In such a case, the function c_A_(y) is constant. To clearly find out the relation between the partial light absorption and the overall reactor efficiency, it is reasonable to assume that the transverse mass dispersion would not delimit the process rate. In such a case, the molar photon flux on the catalyst surface can be easily found as a solution of Equation (5), where the c_A_(y) is constant. The final solution I_a_ = I(y = 0) has the following form:I_a_ = I_0_e^−δεc^_A_,(7)

The substitution of Equation (7) into the kinetic model (4) leads towards the form of Equation (8) which will be used in this simulation.

r_A_ = φI_0_Kc_A_e^−δεc^_A_,(8)

## 2.2. CSTR Model

The first mathematical model used in this article focuses on the description of the relation between the partial light absorption and the overall reaction rate. Therefore, it is reasonable to perform the case of the simplest hydrodynamics. Such conditions may be represented by a CSTR reactor. The schematic drawing of such a reactor is presented in Figure 1a. The control volume for this model is the total liquid volume in the reactor. The concentration of the substrate flowing through the reactor decreases due to the reaction which occurs in the catalyst surface. Therefore, the model is represented by Equation (9).
V dc_A_/dt = F_V_c_A0_ − F_V_c_A_ − Sr_A_,(9)

Substitution of Equation (8) to Equation (9) leads to the form of Equation (10).
V dc_A_/dt = F_V_c_A0_ − F_V_c_A_ − SφI_0_Kc_A_e^−δεc^_A_,(10)

Division two-sided of Equation (10) by the volume V:dc_A_/dt = F_V_/Vc_A0_ − F_V_/V_cA_ − S/VφI0Kc_A_e^−δεc^_A_,(11)

In the next step, I introduced to the model dimensionless variables: the total residence time of liquid in the reactor volume τ = V/F_V_ and the degree of the substrate conversion α = (c_A0_ − c_A_)/c_A0_. I also simplified ratio S/V to 1/δ since the catalyst is deposited on the flat surface, on the tank bottom.
dα/dt = −α/τ − 1/δφI_0_K(1 − α)e^−δεc^_A0_^(1−α)^,(12)

In the steady state, the left side of Equation (12) is equal to 0. Thus, the model in the steady state is represented by Equation (13).
0 = −α/τ − 1/δφI_0_K(1 − α)e^−δεc^_A0_^(1−α)^,(13)

We can introduce now dimensionless numbers to get the model in more general form:(1)the Damköhler number as Equation (14):
Da = τφI_0_K/δ,(14)

By the Damköhler number, the process set-up conditions can be described. The parameter considers both the catalyst properties (Kφ), the reactor irradiation conditions (I_0_) and the reactor scale (τ/δ). From the classical point of view, the proposed form of Equation (13) is referred to the case if the degraded compound does not absorb the radiation. This is an intended approach. In this paper, we focus on the partial light absorption by the liquid film and it is reasonable to represent the optical properties of the liquid film and all others by two separable parameters. Therefore, the second parameter describes the optical properties of the degrading solution and it is called absorbance:(2)the dimensionless expression in the exponent is represented by Equation (15). Truly, it is the absorbance of the liquid film in the case when in the liquid tank the degradation process did not occur. In this paper, the dimensionless parameter κ is shortly named as the absorbance. Computed values of k for selected industrial dyes are presented in Table 1.

κ = δεc_A0_,(15)

The implementation of quantities (14) and (15) into the model Equation (13) leading to the form of Equation (16).
0 = −α − Da(1 − α)e^−κ(1−α)^,(16)

## 2.3. Model with Partial Axial Mixing

The model presented in the last paragraph is related to ideal mixing conditions. Practically, it is not representative to the large-scale case, where the length of the reacting plate should be not too short to maintain ideal mixing conditions. For such a case, the plug-flow model should be more suitable. On the other hand, to reach the proper substrate conversion degree, the total residence time should be in minutes. Therefore, it is reasonable to take into an account also the partial longitudinal mass mixing in the liquid film. The schematic drawing of such a reactor is presented in Figure 1b. The control volume is the cuboid slice of differential length dl. In such a case, the model can be stated by the modification of Equation (11):dsdl∂c_A_/∂t = Udsdlc_A_ − Udsdlc_A_ − Uds∂c_A_/∂ldl − Dds∂c_A_/∂l + Dds∂c_A_/∂l + Dds ∂/∂l(∂c_A_/∂l)dl − sdlI_0_Kφc_A_e^(−δεc_A_)^(17)

Equation (17) should be completed by three boundary conditions (18)–(20) which are based on well-known Danckwert’s conditions:c_A_(t = 0,l) = c_A0_,(18)
c_A_(t,l = 0) − D/Udc_A_/dl(t,l = 0) = c_A0_,(19)
dc_A_(t,l = L)/dt = 0,(20)

The simplification of Equation (17) leads to the form represented by Equation (21).
∂c_A_/∂t= −U∂c_A_/∂l + D∂^2^c_A_/∂l^2^−I_0_Kφc_A_/δe^(−δκc_A_)^,(21)

The total residence time for such a case is defined as a ratio of the reactor length and the liquid film velocity (τ = L/U). This quantity, the substrate conversion degree and others defined by Equations (14) and (15) had been attached to the model (21). Additionally, the dimensionless length has been performed as follows: x = l/L. Finally, the model has been stated by the set of Equation (22)–(25):∂α/∂t= Dτ/L^2^∂^2α^/∂x^2^ − ∂α/∂x + Da(1 − α)e^(−κ(1−α))^,(22)
α(t = 0,x) = 0,(23)
−Dτ/L^2^∂α/∂x(t,x = 0) + α(t,x = 0) = 0,(24)
dα(t,x = 1)/dt = 0,(25)

The ratio L^2^/(Dτ) is known as the longitudinal mass Peclet number, Pe. Therefore, the model in the steady state is represented by the set of Equations (26)–(28):1/Pe d^2α^/dx^2^ − dα/dx + Da(1 − α)e^(−κ(1−α))^ = 0,(26)
dα(x = L)/dx = 0,(27)
−1/Pedα/dx(x = 0) + α(x = 0) = 0,(28)

## 3. Simulation Set-Up

Numerical experiments have been provided both for the ideal-mixing model and the model with longitudinal mass dispersion by using model (26)–(28) to show the reactor properties for two different hydrodynamical regimes.

To find branches of steady states, the software for continuation and bifurcation AUTO-07P has been employed. The obtained branches of steady states describe the region of multiple steady states. It is provided by two-parameter continuation which leads to obtaining the catastrophic plot. The chosen simulation results have been presented in graphical form.

Simulations of the dynamical start-up of the reactor can be based on model (12). Two different start-up scenarios were considered. The first scenario assumes that in the beginning reactor was wet by the feed, whereas in the second case the reactor was wet by pure water. Thus, two other initial conditions have been used, respectively α(t = 0) = 0 and α(t = 0) = 1. The integration of the model (12) has been realized by use of the scientific python library SciPy adapted to the python interpreter named Anaconda.

Solutions of model (26)–(28) have been managed also by AUTO-07P BVP solver. The obtained branches of steady states and catastrophic plots for chosen values of Peclet numbers have been presented. To evaluate the reactor efficiency, I described the substrate conversion degree in the outlet stream which is equal to α_e_ = α(x = 1). To illustrate the influence of the partial absorption of the radiation, I introduced the average reaction rate, which is the mean reaction rate in the reactor volume. This quantity can be calculated by Equation (29):r_avg_= integral (from x = 0 to x = 1) ((1 − α(x))e^−κ(1−α(x))^dx),(29)

## 4. Results

Representative branches of steady states are shown in Figure 2. Presented results show the occurrence of the triple steady-state region when absorbances κ are over 4. Thus, only significantly high light absorption results in occurrence in the multiple steady states appear in the fixed-bed photocatalytic reactors. However, the phenomenon of multiple steady states is combined with the optical properties of the liquid film. As we can see, the branch of steady states for colorless compounds (κ = 0) has no limit point. It is obvious, since when κ = 0, the steady state model (16) becomes linear and it always has one solution. Further numerical experiments also confirm the presented thesis.

In Figure 2b, the same branches are shown. Computed substrate conversion degree values were converted to the degree of the light absorption by using the following formula:I/I_0_ = e_−_^κ(1−α)^,(30)

These branches explain the reason of multiple steady states occurrence. The reactor efficiency is limited by the light incidence in the catalyst surface. On the other hand, the light absorption by the liquid film reduces the light intensity on the catalyst film. Therefore, the light absorption by the liquid film also reduces the light intensity on the catalyst film and, in consequence, the light reaching the catalyst surface has significantly lower intensity in comparison to those reaching the liquid-free surface. In such a case, we can say that a *light barrier* occurs. However, if the concentration of colored compound will be decreased due to the reaction, the light absorption is lower than without process. Therefore, it is possible that the incidence on the catalyst surface will be higher and the reaction rate also. In such a case, the reactor may be stably operated and is reaching the higher substrate conversion degree. The phenomenon may be called a *light barrier breakage*.

To show the multiple steady states region, it is possible to continue a chosen limit point by two parameters. Figure 3 presents a result of the two-parameter continuation, called the catastrophic plot. Figure 3 shows that the triple steady states region is limited by a cusp bifurcation point. The multiple steady states occurrence is limited both by optical properties of the liquid film and the reactor operation conditions.

Apart from the presented results, there may be a preferred solution to operate the process without the existing *light barrier*. In order to perform such a process, the corresponding start-up should be provided. In this paper, I analyzed two different start-up scenarios. The first pathway starts from wetting the catalytic surface by the reactor feed, whereas in the second case the catalytic surface would be wet by pure water. The dynamical simulation of purposed scenarios is shown in Figure 4. Results show that the start-up scenario has a strong influence on the reached steady state value. If the process is started without previous wetting the catalytic plate by the pure water (scenario 1), the *light barrier* will occur and the reactor will reach the lower efficiency. Otherwise (scenario 2), the *light barrier breakage* will happen and the reactor operates on higher substrate conversion degree values. It could be noticed that both scenarios finish in the same steady state, when the process parameters are in the singular steady states region (Figure 4b). Therefore, even a slight change in a parameter value may affect the distinct reduction of a process efficiency.

The other part of the numerical study was combined with the influence of the hydrodynamics for obtained degree of the substrate conversion. Representative simulations results are below.

Figure 5 shows the catastrophic plot shape change when the mass axial Peclet number increases. The change of the reactor hydrodynamics results in shifting the multiple-steady-states region for higher Damköhler numbers and absorbances values. Therefore, the change of the liquid flow regime results in the inability to break the *light barrier* and reach the higher degree of the substrate conversion in the outlet stream. In the case of ideal mixing reactors, the *light barrier breakage* occurs in all the reactor volume because it is followed by the uniform reduction of the degraded compound concentration value in the entire volume of the reactor. On the other hand, in the case of a plug-flow regime, the concentration decreases together with the axial coordinate, so that the *light barrier* effect occurs in the initial section of the plate even when the total residence time is significantly higher.

The further illustration of the influence of the *light barrier* on the overall reactor efficiency is presented in Figure 6. The figure presents branches of steady states versus Peclet number, which represents the diffuse mixing in the liquid stream. When the light absorption by the degraded compound has no influence on the reactor efficiency (κ = 0), the simulation results are typical as in the case of the isotherm tubular reactor with axial mixing. The increase of the Peclet number results in the increase of the substrate conversion degree, similarly, the reactor efficiency increases. However, if the absorbance κ is higher, the highest substrate conversion degree will fall in the case of the ideal mixing in the liquid stream. Thus, the *light barrier* changes the operational property of the reactor. Figure 6 shows that the axial mixing delimits the *light barrier*, whereas the clear plug-flow expands the phenomenon. In such a case, the reactor efficiency has the local maximum and then decreases.

## 5. Discussion

The presented simulations results lead to novel knowledge about photocatalytic reactors. It could be noticed that model, represented by Equation (14), is similar to non-isothermal CSTR reactor with Arrhenius’s kinetics. Properties of such a case have been already well known [24]. However, the simulations bring novel conclusions in physical aspects. First of all, the light absorption by the reagent solution in the case of photodegradation of colored compounds should be taken into an account in every photocatalytic experiment and computation since it may have strong influence on the process efficiency. Most of the experimental works are managed by the simplified pseudo-first-order kinetics models and they do not examine the partial light absorption even then, when the absorption spectrum suggests that the radiation may be partially absorbed before being absorbed by the catalyst [25,26,27,28]. Such experiments may result in untrue kinetic constants or models and consequently lead to incorrect conclusions in further simulations. The solution of simple model (16) exhibits that the substrate conversion degree is related to the optical properties of reactive substrate. Moreover, the feedback relation between the reaction rate and the liquid layer absorbance takes place. So even in a simple experiment, in a tank reactor with ideal mixing the mean reaction rate should be combined with a simple radiation balance (i.e., based on Burger’s law). Otherwise, experimental results are incomplete. The *light barrier breakage*, proposed by the author, suggests that the revision of experimental works may benefit with the increasing the reactor efficiency. Such conclusions may come from this simple simulation showing the significant substrate conversion degree even for the same operational conditions, but with a changed start-up scenario (Figure 4).

The simulation results present simple preposition of increasing the reactor efficiency whether it operates in triple steady-states regions. In the case of comparable models, for example non-isothermal reactors with Arrhenius kinetics, the heat generated during the process has an influence on reactor operation instability. Therefore, the process requires complex control procedure and dedicated heat-exchanges system. In contrast to the mentioned case, the increase of the reaction rate delimits further light absorption, so that the reactor operates like a closed-loop system. The control system does not need to be complex. Nevertheless, the multiple steady states occurrence should be taken into an account in the installation design. The partial light absorption by the degraded compound have also an influence on the operational reactor properties for different hydrodynamic regimes. In comparison to the classical isotheral reactor model, the axial mass dispersion in the liquid stream results in the decrease of the substrate conversion degree. In the case of a photocatalytic reactor, the axial mixing of the substrate allows to delimit the *light barrier* in the reactor volume. Therefore, taking into account the partial light absorption in the reactor model results in reversing of the dependence between Peclet number and the reactor efficiency. Simulations shows the possibility of the complex relation of the diffuse axial mixing and the light absorption on the overall reactor efficiency even when the multiple steady states are not taking place. Therefore, taking into an account the relation between the reactor hydrodynamics and the light absorption balance has a key role in projecting of photocatalytic reactors. It could be noticed that such phenomena may occur also in reactors with suspended nanoparticles. Moreover, the relation between the partial absorption of the radiation by the degraded compound and catalytic nanoparticles may be more complex. From that point of view, all experiments managed in photocatalytic degradation of compound which may partially absorb the radiation should be again reviewed.

The obtained solutions have shown the unknown aspect of photodegradation. The *light barrier* phenomenon has a strong influence on the reactor operation and should be taken into an account both in experimental works and in reactor designing. Multiple steady states have been observed even in a simple-structured reactor. However, the *light barrier* does not eliminate the process to being employed in the industry. The presented simulation results show that it is possible to safely operate the *light barrier breakage* and cause an increasing of the reactor efficiency for even several times. This paper shows tends in general abstract cases. Therefore, the *light barrier* occurrence phenomenon should be experimentally confirmed in future. Nevertheless, the obtained solutions suggest that the possibility of its affecting on the overall reactor efficiency exists.

Presented here, the scheme of the multiple steady states analysis dedicated to real technological cases in photocatalytic reactors should be carefully adapted for every single construction of the reactor. First of all, it comes from topological differences between the physics of light and mass transfer. It should be taken into an account that photons flow has a different nature than the mass transfer in reactors. Secondly, the form of catalyst determines the path of the light penetration truth the reacting solution. This applies not only for the difference between slurry and immobilized-bed reactors, but also the form of immobilization and the distance between the light source and the place of catalytic deposition which directly affects the optical layer thickness. Last but not least, the hydrodynamical regime of the liquid stream inside the reactor should be taken into an account. All of abovementioned properties directly determine the operational parameter range (Da, Pe and κ may be not sufficient to describe a separable case) for which the simulation results are referred to the real, technological case. Nevertheless, this paper shows the general trends, that the *light barrier* may occur, and found out the general hint as to how to reach the *light barrier breakage*.

## Figures and Tables

**Figure 1 molecules-26-03804-f001:**
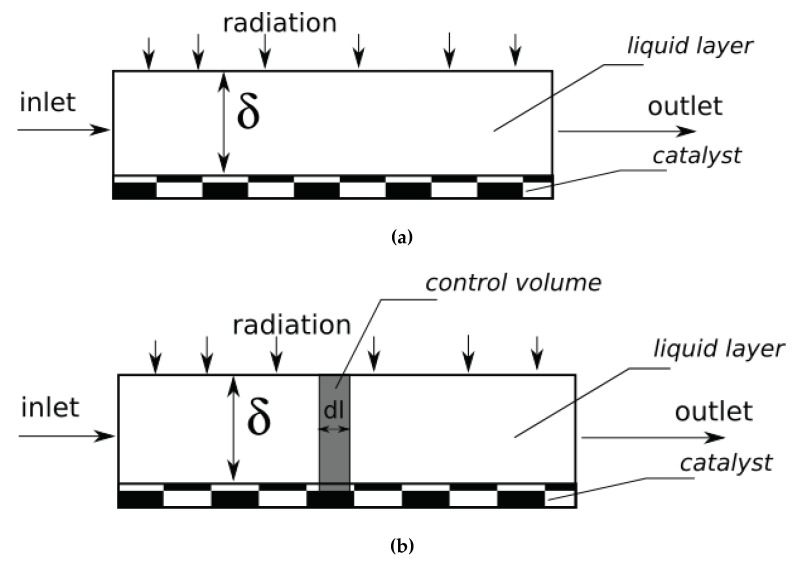
The schematic drawing of the modeled reactor: (**a**) the CSTR approach where the control volume is the total liquid volume in the reactor; (**b**) the control volume is symbolically shown as a gray rectangle.

**Figure 2 molecules-26-03804-f002:**
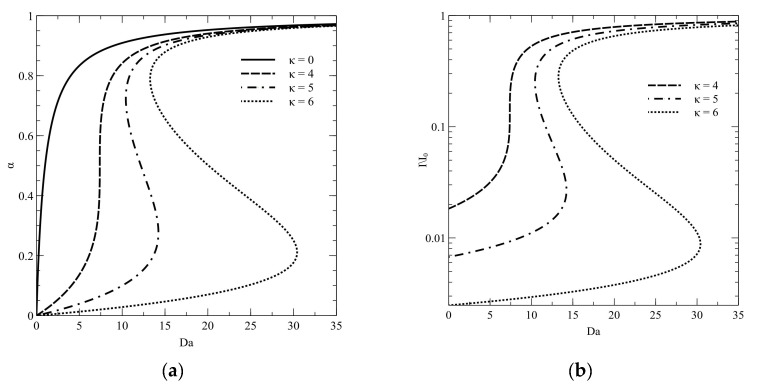
Branches of steady states for different absorbance of the liquid layer: (**a**) the substrate conversion degree; (**b**) relative light incident on the catalyst surface.

**Figure 3 molecules-26-03804-f003:**
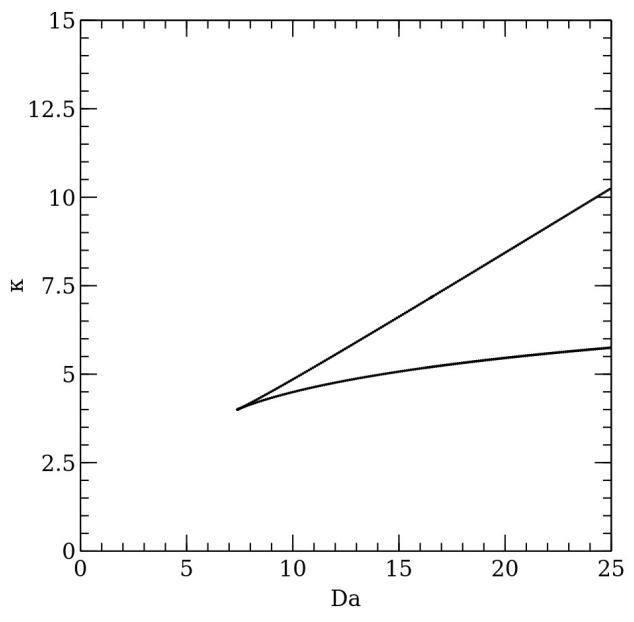
The catastrophic plot—the two-parametric relation. For parameters values between the curve, triple steady states occur. Otherwise, singular steady states take place.

**Figure 4 molecules-26-03804-f004:**
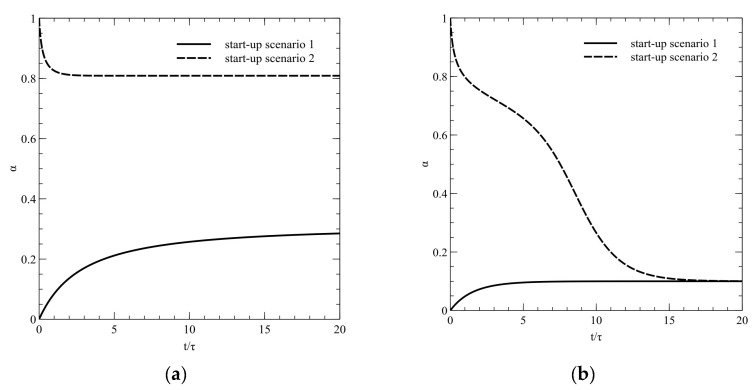
Dynamical simulations of reactor start-up for Da = 10. The absorbance was equal to: (**a**) 4.5; (**b**) 5, respectively.

**Figure 5 molecules-26-03804-f005:**
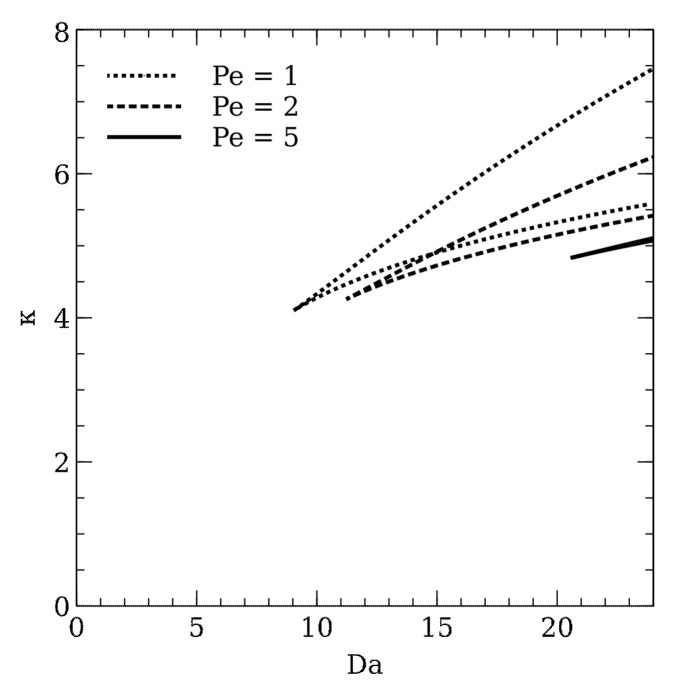
Catastrophic plots for different Pe numbers. The change of the cusp bifurcation point occurs.

**Figure 6 molecules-26-03804-f006:**
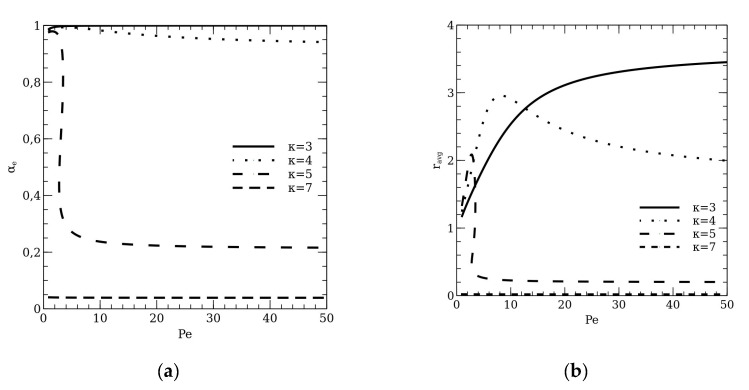
The branches of steady states versus the Peclet number for different absorbances. Two different quantities have been presented in this figure: (**a**) the substrate conversion degree in the outlet stream; (**b**) the mean dimensionless reaction rate. The Damköhler number for these simulations is set on Da = 20.

**Table 1 molecules-26-03804-t001:** Computed κ values for 10 ppm water solutions of industrial dyes penetrated by a monochromatic radiation wavelength λ = 254 nm, corresponding to high-pressure mercury lamp maximum spectra. The height of the optical layer δ is equal to 5 cm.

Compound	Methylene Blue	Crystal Violet	Auramine O	Malachite Green	Methyl Orange
κ values for 10 ppm	4.5	17.9	3.1	5.4	7.8
water solutions					

## Data Availability

This study is not report any data.

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
