# Peer review of "Multiple Steady States in the Photocatalytic Reactor for Colored Compounds Degradation"

_molecules, 2021, doi:10.3390/molecules26133804_

Round 1
Reviewer 1 Report
This is an interesting and useful manuscript that readers of Molecules would find useful. The mechanistic proposal is based one extensive mathematical modeling without any clear experimental support. Generally I think any mathematical model without experimental support would not be useful. Hope the author can find a collaborator to verify the predictions. If not this article will lie buried in Molecules.
Author Response
Thank you for your review. I send the corrected version of manuscript where, in my opinion, the language level is improved.
In future I planned to lead the experimental verification of the mathematical models. The reactor has been already under construction.
Reviewer 2 Report
Please see attached comnets on the document.

Author Response
Dear reviewer,
Thank you for your detailed revision. Your work showed me that my manuscript was not clearly written. I had been tried to apply all your tips you provide me. This results in complete rewritting some paragraphs. The correction was too general to underline all changes.
Following your advices, in the introduction I add the general review about the multiple steady states in chemical reactors. Then I tried to make a connection between the classical case and this research.
Your make me realize about the chaoic form of the mathematical model derivation. Therefore I parted that paragraph into the 3 separated paragraphs. In the first I disscused in details about the kinetics in the case of degradation of colored wastewater contaminant. The second focus on the CSTR model, which is general and the third manage about the model with partial axial mixing of the liquid stream.
I thinking a lot after your proposition about the potential refference of simulations results to the realistic case. I come to the conclusion that your (neighter my) expectations are not easily to apply. This is because of the different nature between radiation and mass flow. The modeling of packed-bed reactor would require much more complex mathematical operations. Differences comes from the following reasons:
1) different catalyst distribution in the reacting channel. The catalyst present on a catalyst support like glass spheres or silica beds is distribute in all the volume of the reactor, whereas the catalyst in such tank reactor is located on the bottom. Therefore the radiation term in this case will be more complex or even require separate differential equation with respect to the tube diameter to solve;
2) the light rays structure is different than in the case of a cuboid tank. We cannot apply just cylindrical coordinates with respect to the radiation. The radiative transport has a different nature than the convective one.
For those reasons I think that the model with respect to the light transport will have more complex form whereas I found this work as a presentation of one phenomenon - the interaction between the partial light absorption and the kinetic equation. If the presented case will be more complex then I cannot clearly show the altitude between mentioned factors. However, your proposition about an application the analysis of multiple steady states for one, technological case I found as a great idea, nevertheless I found the different function of this paper. I tried to emphasize it in the discussion paragraph.
Potencially, the multiple steady states structure in the reactor with taking into account the reactor hydrodynamics, topological differences and the catalyst distribution may be even more developed than such a simple case. On the other hand, I think that the description of the presented concept is more clear when I used more abstract example.
Once again I am grateful for all your advices. In my opinion it helps me to improve my manuscript and also this is a big piece of knowlegde bomb for preparing my further works.